# A human-specific motif facilitates CARD8 inflammasome activation after HIV-1 infection

**Jessie Kulsuptrakul[1], Elizabeth A Turcotte[2], Michael Emerman[3]\*, Patrick S Mitchell[4]\***

[1]Molecular and Cellular Biology Graduate Program, University of Washington, Seattle, United States; [2]Division of Immunology and Pathogenesis, University of California, Berkeley, Berkeley, United States; [3]Divisions of Human Biology and Basic Sciences, Fred Hutchinson Cancer Center, Seattle, United States; [4]Department of Microbiology, University of Washington, Seattle, United States

**Abstract** Inflammasomes are cytosolic innate immune complexes that assemble upon detection of diverse pathogen-associated cues and play a critical role in host defense and inflammatory pathogenesis. Here, we find that the human inflammasome-forming sensor CARD8 senses HIV-1 infection via site-specific cleavage of the CARD8 N-terminus by the HIV protease (HIV-1$^{PR}$). HIV-1$^{PR}$ cleavage of CARD8 induces pyroptotic cell death and the release of pro-inflammatory cytokines from infected cells, processes regulated by Toll-like receptor stimulation prior to viral infection. In acutely infected cells, CARD8 senses the activity of both de novo translated HIV-1$^{PR}$ and packaged HIV-1$^{PR}$ that is released from the incoming virion. Moreover, our evolutionary analyses reveal that the HIV-1$^{PR}$ cleavage site in human CARD8 arose after the divergence of chimpanzees and humans. Although chimpanzee CARD8 does not recognize proteases from HIV or simian immunodeficiency viruses from chimpanzees (SIVcpz), SIVcpz does cleave human CARD8, suggesting that SIVcpz was poised to activate the human CARD8 inflammasome prior to its cross-species transmission into humans. Our findings suggest a unique role for CARD8 inflammasome activation in response to lentiviral infection of humans.

**\*For correspondence:**
memerman@fredhutch.org (ME);
psmitche@uw.edu (PSM)

**Competing interest:** The authors declare that no competing interests exist.

## Editor's evaluation

Kulsuptrakul and colleagues provide convincing evidence that the human inflammasome-forming sensor CARD8 contains a specific F-F motif that allows cleavage by the proteases of HIV-1 and emerged after separation of chimpanzees and humans. In comparison, CARD8 proteins from non-human primates contain changes in this motif and are largely resistant to proteolytic activation. These important findings suggest a potential role of CARD8 cleavage and inflammasome activation in primate lentiviral pathogenesis.

## Introduction

One of the primary selective pressures that shape viral adaptation to a new host, as well as tolerance to persistent infections, is the innate immune system (**Daugherty and Malik, 2012**; **Parrish et al., 2008**). One class of innate immune sensors forms cytosolic immune complexes called inflammasomes, which initiate inflammatory signaling upon pathogen detection or cellular stress (**Broz and Dixit, 2016**). Inflammasome activation is critical for host defense against a wide range of pathogens; however, auto-activating mutations in inflammasome-forming sensors can also initiate inflammatory

pathogenesis that drives autoinflammatory and autoimmune disorders (*Steiner et al., 2018*; *Taaba-zuing et al., 2020*).

The inflammasome-forming sensor caspase recruitment domain-containing protein 8 (CARD8) consists of a disordered N-terminus, a function-to-find domain (FIIND), and a caspase activation and recruitment domain (CARD) (*Taabazuing et al., 2020*). The FIIND, comprised of ZU5 and UPA subdomains, undergoes self-cleavage resulting in two non-covalently associated fragments (*D'Osualdo et al., 2011*; *Taabazuing et al., 2020*). Proteasome-dependent degradation of the N-terminus leads to the release and assembly of the C-terminal UPA-CARD, serving as a platform for the recruitment and activation of caspase-1 (CASP1). Activated CASP1 initiates a lytic, programmed cell death called pyroptosis and the release of pro-inflammatory cytokines including interleukin (IL)-1β and IL-18 (*Broz and Dixit, 2016*; *Fink and Cookson, 2005*). To prevent aberrant release of the UPA-CARD, the dipeptidyl peptidases 8 and 9 (DPP8/9) form an inhibitory complex with CARD8 (*Sharif et al., 2021*).

The CARD8 inflammasome can be activated by several triggers. For example, disruptions to protein homeostasis, including direct (e.g. Val-boroPro) and indirect (e.g. CQ31) inhibition of DPP8/9, cause CARD8 inflammasome activation (*Johnson et al., 2020*; *Johnson et al., 2018*; *Rao et al., 2022*). Several recent examples also highlight CARD8 inflammasome activation in response to pathogens (*Nadkarni et al., 2022*; *Tsu et al., 2023*), including via its recognition of the enzymatic activity of the HIV-1 protease (HIV-1$^{PR}$) (*Wang et al., 2021*). For example, treatment of HIV-1 latently infected cells with certain nonnucleoside reverse transcriptase inhibitors (NNRTIs), including efavirenz (*Wang et al., 2021*) or doravirine-like analogs including the pyrimidines Pyr01 (*Balibar et al., 2023*), enforce the cytosolic dimerization of the HIV-1$^{PR}$ and results in CARD8 inflammasome activation in primary CD4+ T cells and humanized mouse models (*Clark et al., 2022*). HIV-1$^{PR}$ cleavage of the N-terminus of CARD8 causes proteasome-dependent degradation of the CARD8 N-terminal fragment (*Wang et al., 2021*). This 'functional degradation' liberates the UPA-CARD fragment for inflammasome assembly and activation, analogous to viral protease sensing by the inflammasome-forming sensor NLRP1 (*Robinson et al., 2020*; *Sandstrom et al., 2019*; *Tsu et al., 2021a*; *Tsu et al., 2021b*; *Planès et al., 2022*) in which the N-terminus of CARD8 functions as a molecular 'tripwire' to sense and respond to the enzymatic activity of HIV-1$^{PR}$ and other viral proteases (*Castro and Daugherty, 2023*; *Nadkarni et al., 2022*; *Tsu et al., 2023*).

Here, we report that CARD8 can also sense acute HIV-1 infection via the detection of HIV-1$^{PR}$ activity. We find that priming of target cells via Toll-like receptor (TLR) agonists prior to HIV-1 challenge enhances CARD8-dependent cell death and is required for IL-1β secretion. Our evolution-guided studies reveal that CARD8 sensing of HIV-1 and other simian lentiviruses is dependent on a F59-F60 motif in human CARD8 that permits its sensing of HIV-1$^{PR}$. This motif is absent in other primates that serve as reservoirs of simian immunodeficiency viruses (SIVs), and although both HIV-1$^{PR}$ and SIVcpz$^{PR}$ cleave and activate human CARD8, we find that neither are sensed by chimpanzee CARD8. Thus, our study reveals that the CARD8 inflammasome functions in the innate immune detection of HIV-1 replication. Moreover, our findings suggest that the evolution of the F59-F60 motif in humans gave rise to a human-specific host-virus interaction following the spillover of SIVcpz into humans, which may uniquely shape human innate immune responses to lentiviral infection.

## Results

### A human-specific motif allows CARD8 to detect protease activity from multiple HIV strains

The HIV-1 protease (HIV-1$^{PR}$) cleaves human CARD8 between phenylalanine (F) 59 (P1) and F60 (P1') (*Figure 1A*; *Wang et al., 2021*). While the amino acid P1 site, F59, is invariant among hominoids, gibbons, and Old World monkeys, only human CARD8 has a phenylalanine at the P1' site, F60 (*Figure 1A*). The F59-F60 motif therefore must have arisen in the human lineage after the most recent common ancestor with chimpanzees and bonobos. The F59-F60 motif is also present in *Homo neanderthalensis* (i.e. Neanderthal) CARD8 (*Figure 1A*), conservatively dating its emergence within the last million years (*Green et al., 2010*).

In order to assess the significance of the human CARD8 F59-F60 motif, we established conditions required for HIV$^{PR}$ cleavage of CARD8 by co-expression of CARD8 and proviruses from two HIV-1 group M proviruses (HIV-1$_{LAI}$ subtype B and HIV-1$_{Q23}$ subtype A) as well as an HIV-2 isolate, HIV-2$_{ROD}$.

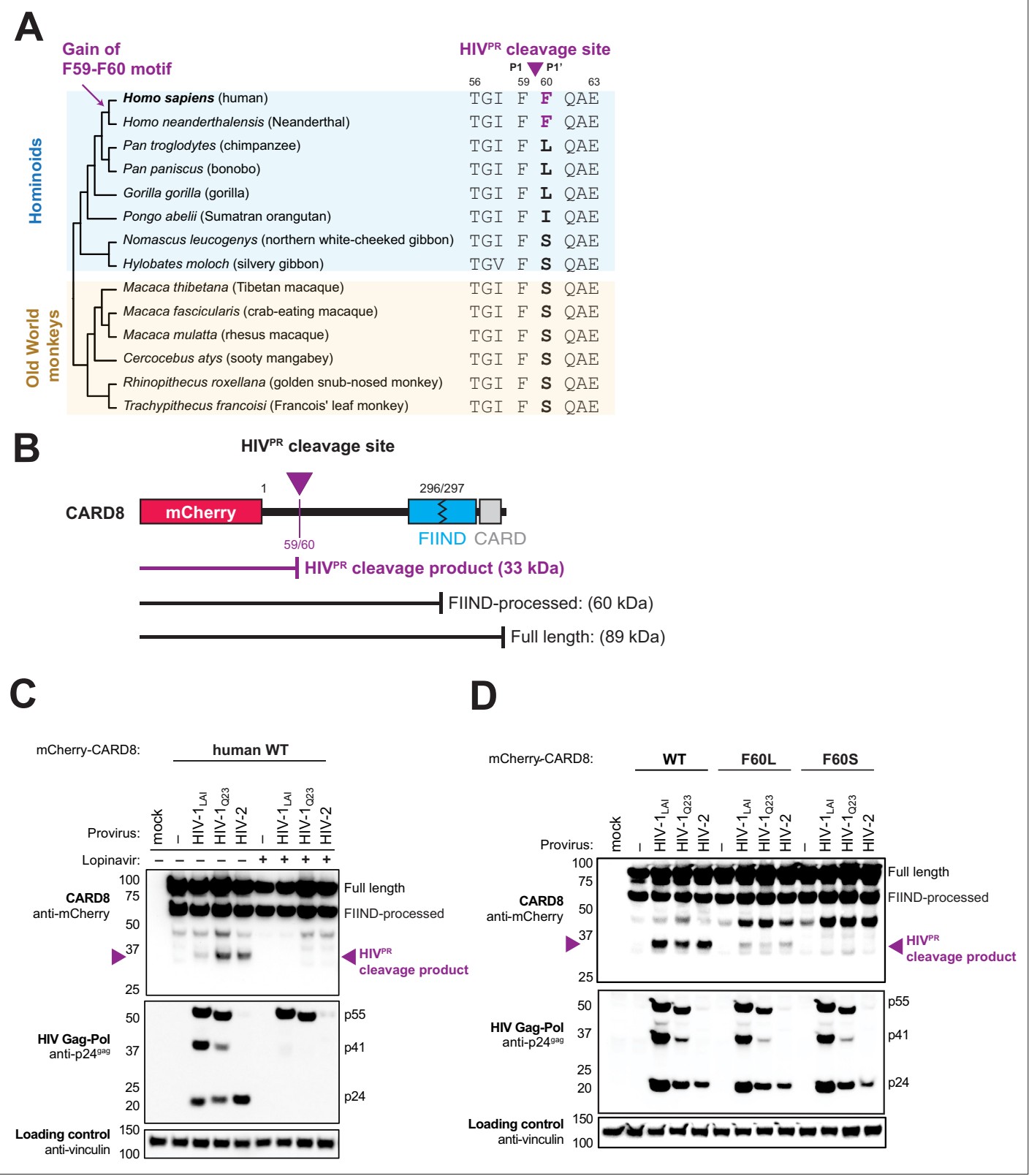

**Figure 1.** The F59-F60 motif allows human CARD8 to detect protease activity from multiple HIV strs. (**A**) Phylogenetic alignment of primate CARD8 protein sequences. The HIV protease (HIV$^{PR}$) cleavage site is indicated by a purple triangle between F59 (P1) and F60 (P1'). Numbering is based on human CARD8. (**B**) Depiction of the mCherry-CARD8 used in cleavage assays in (**C and D**). The predicted molecular weights (kDa) for full-length, FIIND-processed, or HIV$^{PR}$ cleavage products are indicated. FIIND, function-to-find domain; CARD, caspase activation and recruitment domain. (**C**) HEK293T

*Figure 1 continued on next page*

*Figure 1 continued*

cells were transfected with a construct encoding N-terminally mCherry-tagged wildtype (WT) CARD8 and indicated HIV proviral constructs, HIV-1$_{LAI}$, HIV-1$_{Q23}$, or HIV-2$_{ROD}$, in the presence ('+') or absence ('−') of 10 μM lopinavir, an HIV$^{PR}$ inhibitor. *Top:* Immunoblotting for anti-mCherry to detect the mCherry-CARD8 fusion protein. The full-length and FIIND-processed bands are indicated as well as the HIV$^{PR}$ cleavage product. The band at ~45 kDa is the result of cleavage by the 20S proteasome (*Hsiao et al., 2022*). *Middle*: Immunoblotting with an anti-p24$^{gag}$ antibody showing Gag cleavage products p41$^{gag}$ and p24$^{gag}$, and/or full-length Gag, p55$^{gag}$. *Bottom*: Immunoblotting with anti-vinculin as a loading control. (**D**) HEK293T cells were transfected with a construct encoding N-terminally mCherry-tagged WT, F60L, or F60S CARD8 and indicated HIV proviral constructs. Immunoblotting and labeling of the blots as in (**C**).

Indeed, we found that wildtype (WT) human CARD8 with an N-terminal mCherry fusion is cleaved upon transfection of HIV-1 and HIV-2 proviruses, resulting in an ~33 kDa product (*Figure 1B and C*, top blot). The band at ~45 kDa is the result of cleavage by the 20S proteasome and results in a non-functional product (*Hsiao et al., 2022*). Cleavage of CARD8 in these experiments was dependent on the protease encoded by the *Gag-Pol* gene of these proviruses as the HIV$^{PR}$ inhibitor lopinavir (LPV) blocked both Gag processing of p55$^{gag}$ to p41$^{gag}$ and p24$^{gag}$ and CARD8 cleavage (*Figure 1C*, top and middle blot). To evaluate the significance of the amino acid variation at the F60 P1' site of CARD8, we next replaced human CARD8 F60 (WT) with either a leucine (L; found in chimpanzee, bonobo, and gorilla) or a serine (S; found in gibbons and Old World monkeys) (*Figure 1A*). HIV$^{PR}$ cleavage of WT human CARD8 (F60) was much more efficient than cleavage of human CARD8 F60L or F60S (*Figure 1D*), consistent with prior findings that an alanine at position 60 also blocks HIV$^{PR}$ (*Wang et al., 2021*). These results indicate that species-specific variation at position 60 impacts CARD8 recognition of HIV$^{PR}$ activity.

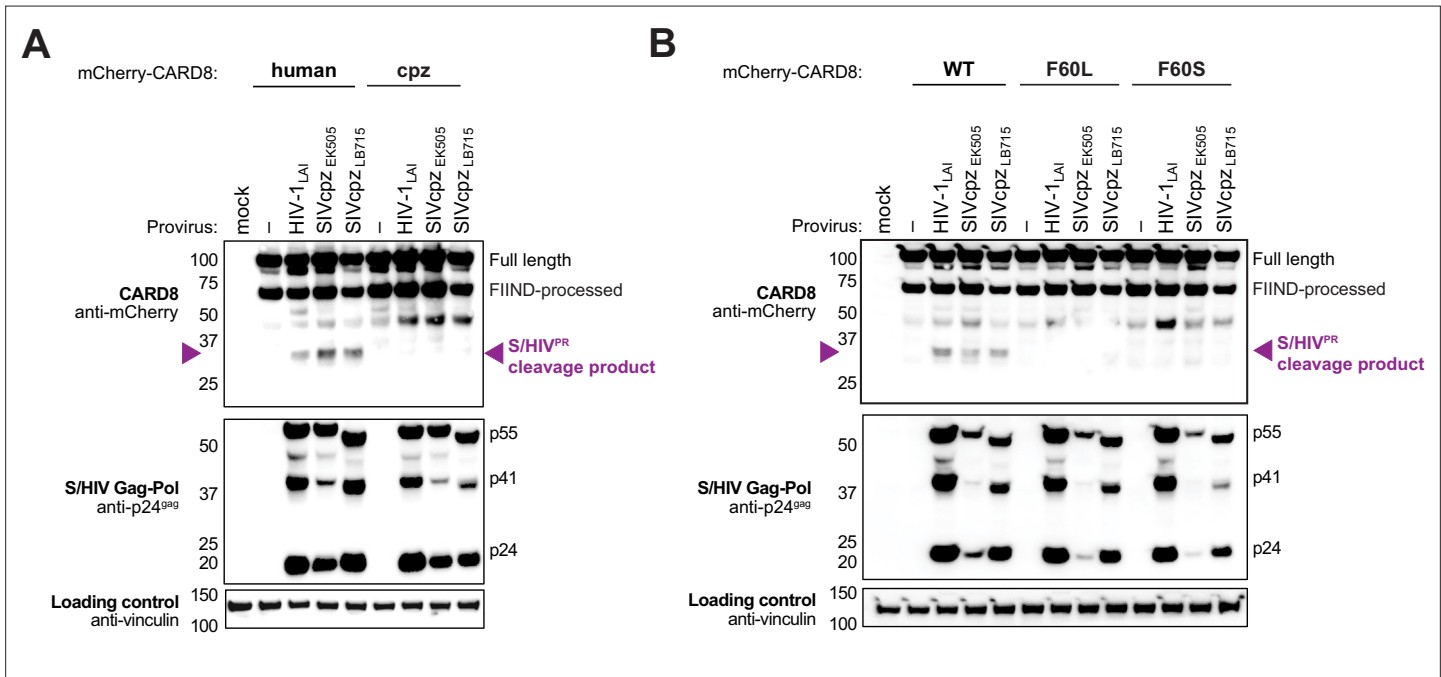

**Figure 2.** Natural variation in CARD8 alters sensing of SIVcpz$^{PR}$ activity. (**A**) HEK293T cells were transfected with a construct encoding N-terminally mCherry-tagged human or chimpanzee (cpz) CARD8 and indicated provirus constructs. Immunoblotting was carried out for CARD8 cleavage, HIV/SIV protease (S/HIV$^{PR}$) activity, and vinculin (loading control) as indicated. The S/HIV$^{PR}$ cleavage product is indicated by a purple triangle. FIIND, function-to-find domain. (**B**) HEK293T cells were transfected with a construct encoding N-terminally mCherry-tagged wildtype (WT), F60L, or F60S CARD8 and indicated proviral constructs. Immunoblotting was performed as in (**A**).

The online version of this article includes the following figure supplement(s) for figure 2:

**Figure supplement 1.** SIVmac cleaves wildtype (WT) human CARD8.

## Natural variation in CARD8 alters sensing of SIVcpz protease activity

We next asked if HIV$^{PR}$ cleavage of CARD8 was an ancestral function of SIVcpz or if that functionality instead emerged following cross-species transmission and adaptation to humans. SIVcpz$_{EK505}$ and SIVcpz$_{LB7}$ represent lineages that gave rise to HIV-1 group N and M viruses, respectively (*Barbian et al., 2015*; *Keele et al., 2006*; *Sharp and Hahn, 2011*). Like HIV-1 and HIV-2 proteases, we found that both SIVcpz proteases (SIVcpz$^{PR}$) cleaved human CARD8 (*Figure 2A*), suggesting that SIVcpz$^{PR}$ had a pre-existing ability to cleave human CARD8 prior to spillover. To deduce whether or not this cleavage is unique to humans, we also tested SIVcpz$^{PR}$ ability to cleave chimpanzee CARD8 (*Figure 2A*) and F60L and F60S human CARD8 variants (*Figure 2B*) and found that none of the other CARD8 variants could be cleaved by SIVcpz$^{PR}$. Moreover, SIVmac$_{239}^{PR}$ also cleaved WT human CARD8, an event that was greatly reduced when tested against the human CARD8 cleavage mutant F60A (*Figure 2—figure supplement 1*). These data suggest that SIVcpz$^{PR}$ was poised to cleave human CARD8 prior to its zoonosis to humans. Moreover, the F59-F60 motif that arose in the human lineage renders human CARD8 uniquely susceptible to cleavage at that position by a broad range of primate lentiviral proteases.

## HIV-1 infection activates the inflammasome in primed THP-1 cells in a CARD8-dependent manner

We next sought to determine the significance of CARD8 cleavage and activation in the context of HIV-1 infection. Treatment with some NNRTIs induces premature Gag-Pol dimerization and HIV-1$^{PR}$ activity (*Figueiredo et al., 2006*; *Trinité et al., 2019*), which was previously shown to be required for CARD8 activation in HIV-1 latently infected cells (*Clark et al., 2022*; *Wang et al., 2021*). However, we observed Gag processing of p55$^{gag}$ to p41$^{gag}$ and p24$^{gag}$ in cytoplasmic lysates of THP-1 cells infected with either WT HIV-1$_{LAI}$ or HIV-1$_{LAI}$ that was pseudotyped with the vesicular stomatitis virus glycoprotein (VSV-g) instead of its own envelope (HIV-1$_{LAI-VSVG}$), consistent with prior studies demonstrating that some HIV-1$^{PR}$ is active in the cytoplasm (*Figure 3A*; *Alvarez et al., 2006*; *Tabler et al., 2022*). To determine if CARD8 inflammasome activation can occur during HIV-1 infection in the absence of small molecule-induced HIV-1$^{PR}$ dimers, we infected the human leukemia monocytic cell line THP-1 at a multiplicity of infection (MOI) <1 and assayed for cell death (*Figure 3B*, left) or IL-1β secretion (*Figure 3B*, right). As a positive control for inflammasome activity, uninfected cells were also treated with VbP, which specifically activates the CARD8 inflammasome in THP-1 cells (*Johnson et al., 2020*). For both HIV-1-infected and VbP-treated THP-1 cells, we observed an increase in cell death compared to mock-infected controls as measured by uptake of the membrane-impermeable dye propidium iodide (PI) (*Figure 3B*, left). However, neither HIV-1 infection nor VbP alone led to an increase in IL-1β levels (*Figure 3B*, right**,** no prime condition), consistent with prior reports (*Ball et al., 2020*; *Linder et al., 2020*). We reasoned that the lack of cytokine production may either be an intrinsic property of CARD8 (*Ball et al., 2020*) or, alternatively, require a signal (e.g. a TLR agonist) to transcriptionally upregulate or 'prime' IL-1β and/or inflammasome components. Thus, we assessed inflammasome activation by HIV-1 infection or VbP treatment with and without pretreating THP-1 cells with agonists of TLR1/2 (Pam3CSK4), TLR7/8 (CL075), TLR8 (TL8-506), or TLR4 (LPS). We found that VbP treatment and HIV-1$_{LAI-VSVG}$ infection induce cell death independent of priming, although TLR agonists did elevate cell death responses in some instances (*Figure 3B* left). In contrast, the release of IL-1β after HIV-1 infection or VbP treatment was entirely dependent on TLR priming (*Figure 3B*, right). Thus, HIV-1 infection alone (i.e. in the absence of molecules causing premature dimerization of Gag-Pol) is sufficient to induce cell death in THP-1 cells, and priming (e.g. via TLR stimulation) is required for HIV-1 infection-induced IL-1β secretion and elevated levels of cell death.

To determine if inflammasome activation upon HIV-1 infection is dependent on CARD8, we generated clonal THP-1 *CARD8* knockout (KO) cells via CRISPR/Cas9. We confirmed the absence of full-length (~62 kDa) and FIIND-processed (~29 kDa) CARD8 in *CARD8* KO THP-1 cell lines by immunoblotting with an antibody specific to the CARD8 C-terminus (*Figure 3—figure supplement 1A*). To functionally test the THP-1 *CARD8* KO cell lines, we primed WT or *CARD8* KO THP-1 cells with Pam3CSK4 then treated with either VbP, which activates the CARD8 inflammasome, or the ionophore nigericin, which specifically activates the NLRP3 inflammasome, and measured cell death and IL-1β secretion. As expected, WT but not *CARD8* KO THP-1 cells responded to VbP, whereas both cell lines underwent cell death and IL-1β secretion in response to nigericin, indicating that the *CARD8* KO

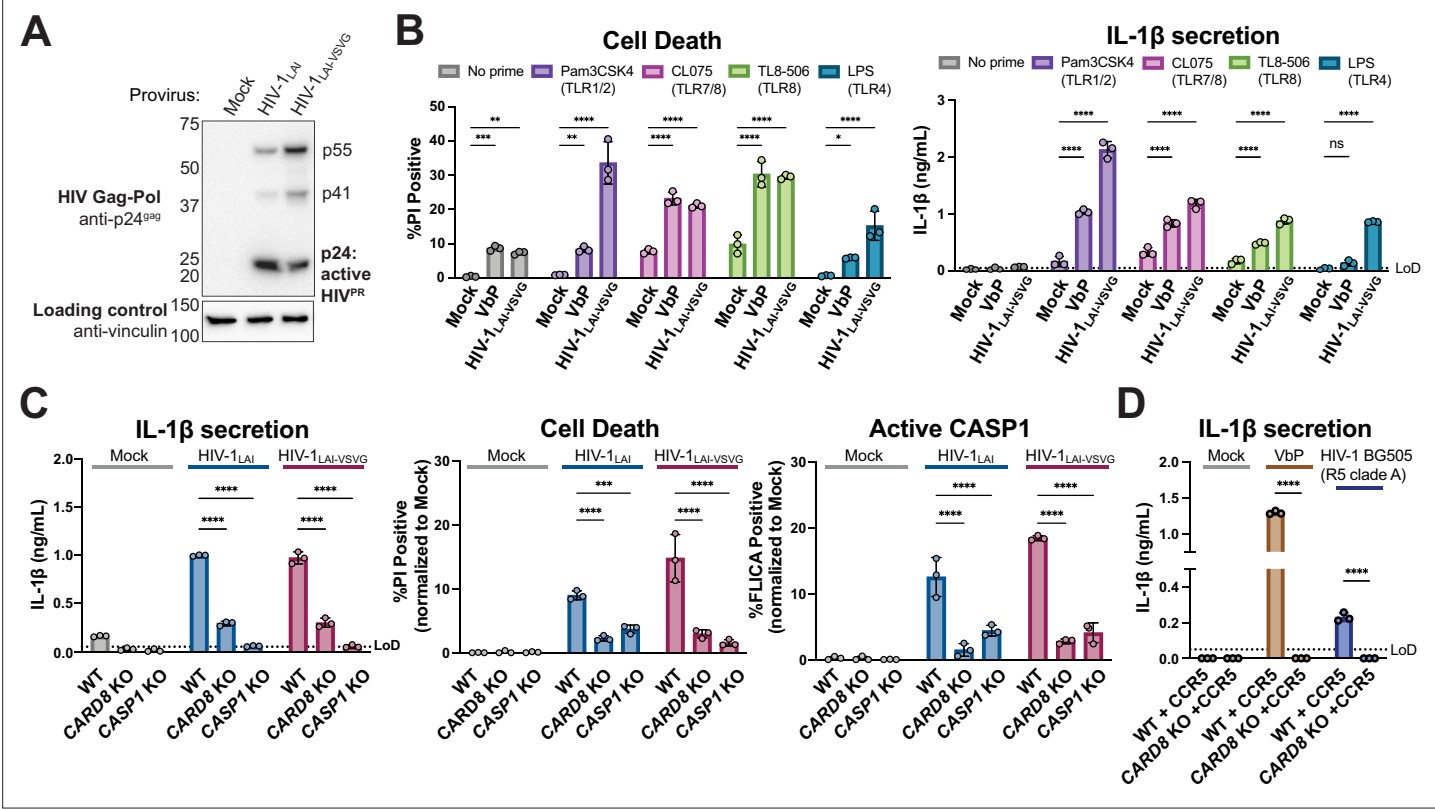

**Figure 3.** HIV-1 infection activates the CARD8 inflammasome in THP-1 cells. (**A**) THP-1 cells were mock infected or infected with HIV-1$_{LAI}$ or HIV-1$_{LAI-VSVG}$, yielding 8% and 53% p24$^{gag}$+ cells after 24 hr, respectively. Immunoblotting using cytoplasmic lysates was carried out for HIV protease (HIV$^{PR}$) activity, and vinculin (loading control) as indicated 24 hr post-infection. (**B**) THP-1 cells were either left unprimed or primed with different Toll-like receptor (TLR) agonists 4–6 hr before treatment with 10 μM VbP or infection with HIV-1$_{LAI-VSVG}$ at a multiplicity of infection (MOI) such that 30–50% were p24$^{gag}$+ after 24 hr. Inflammasome responses were measured 24 hr following VbP treatment or HIV-1 infection. *Left:* Cell death is reported as the percent of propidium iodide (PI) positive cells. *Right:* Interleukin (IL)-1β levels were measured using the IL-1R reporter assay. (**C**) Wildtype (WT), *CARD8* knockout (KO), or caspase-1 (*CASP1*) KO THP-1 lines were primed with Pam3CSK4 then challenged with either HIV-1$_{LAI}$ or HIV-1$_{LAI-VSVG}$ at an MOI such that 30–50% of WT cells were p24$^{gag}$+ after 24 hr. Subsequent inflammasome activation was assayed 24 hr post-infection. *Left and middle*: Cell death and IL-1β levels were measured as in (A). *Right*: Active CASP1 was measured with CASP1-specific FLICA dye. (**D**) WT or *CARD8* KO THP-1s overexpressing CCR5 were primed and treated with 10 μM VbP or infected with HIV-1$_{BG505}$ for 24 hr such that ~30% of cells were p24$^{gag}$+ then probed for inflammasome activation via IL-1β secretion. HIV-1$_{BG505}$ is a CCR5 tropic strain in clade A. The dotted line indicates limit of detection (LoD). Datasets represent mean ± SD (n=3 biological replicates). p-Values were determined by two-way ANOVA with Dunnett's (**B–C**) or Sidak's (**D**) test using GraphPad Prism 9. ns = not significant, *p<0.05,**p<0.01, ***p<0.001, ****p<0.0001.

The online version of this article includes the following source data and figure supplement(s) for figure 3:

**Source data 1.** Tables of source data for propidium iodide uptake, FLICA, and IL-1β secretion.

**Figure supplement 1.** Functional validation of *CARD8* knockout (KO) THP-1 cells.

**Figure supplement 1—source data 1.** Tables of source data for propidium iodide uptake and IL-1β secretion.

THP-1 cells retained responsiveness to other inflammasome agonists (*Figure 3—figure supplement 1B*).

We next infected both WT, *CARD8* KO, or *CASP1* KO THP-1 cells with WT HIV-1$_{LAI}$ or HIV-1$_{LAI-VSVG}$ viruses at an MOI that would give 30–50% infection of WT cells. Similar to our observations with VbP, we found that IL-1β secretion, cell death, and CASP1 activation (as measured by FLICA assay) were significantly reduced in *CARD8* KO versus WT THP-1 cells following HIV-1 infection (*Figure 3B*). Because responses to HIV-1 infection were reduced to a similar level in both *CARD8* KO cells and *CASP1* KO cells, our findings suggest that the inflammasome response to HIV-1 infection in THP-1 cells is primarily dependent on CARD8, but independent of HIV-1 envelope. As different HIV-1 and SIV proviruses were found to cleave human CARD8 after transfection in 293T cells (*Figure 1* and *Figure 2*), we also tested a primary isolate of HIV-1 from a different clade and with a different co-receptor usage

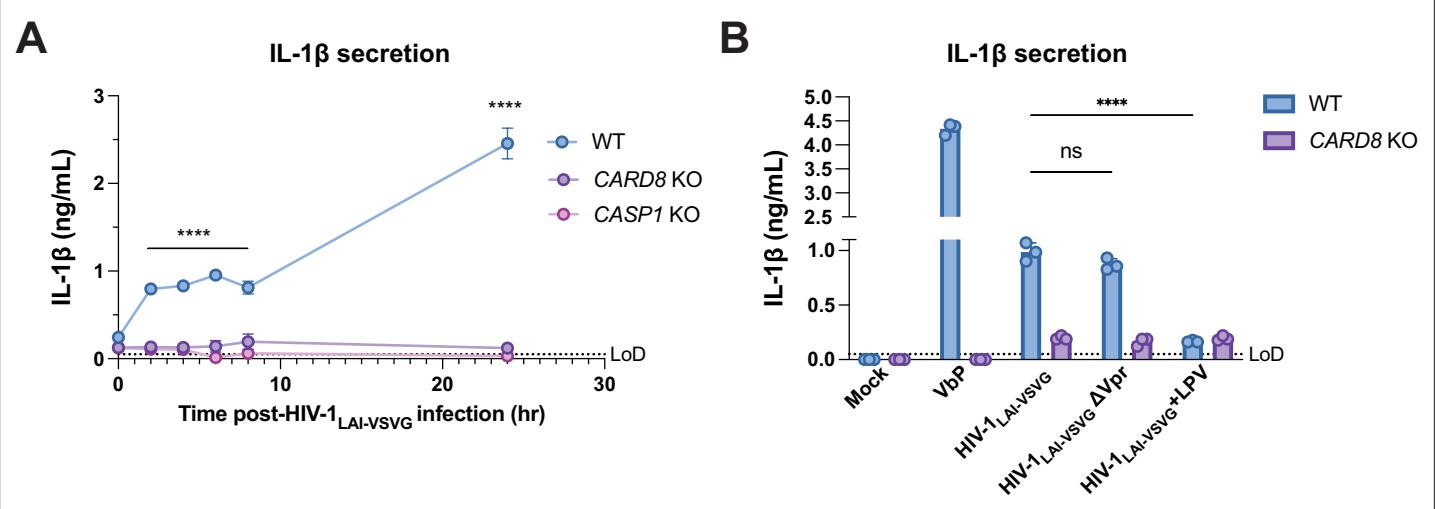

**Figure 4.** Incoming and outgoing HIV$^{PR}$ ar responsible for CARD8 inflammasome activation. (**A**) Wildtype (WT), *CARD8* knockout (KO), or caspase-1 (*CASP1*) KO THP-1 cells were primed overnight with Pam3CSK4 and then infected with HIV-1$_{LAI-VSVG}$. Supernatant was collected at 0, 2, 4, 6, 8, and 24 hr post-infection to measure interleukin (IL)-1β secretion. Cells were infected at viral concentration such that ~70% of cells were p24$^{gag}$-positive after 24 hr. (**B**) WT THP-1 cells primed with Pam3CSK4 were challenged with WT or mutant HIV-1$_{LAI-VSVG}$ or WT virus preincubated in 5 μM lopinavir (LPV) for 30 min prior to infection. HIV-1$_{LAI-VSVG}$ ΔVpr has a frameshift mutation in Vpr. Dotted line indicates limit of detection (LoD). Datasets represent mean ± SD (n=3 biological replicates). p-Values were determined by two-way ANOVA with Dunnett's test using GraphPad Prism 9. ns = not significant, *p<0.05, **p<0.01, ***p<0.001, ****p<0.0001.

The online version of this article includes the following source data for figure 4:

**Source data 1.** Tables of source data for IL-1β secretion.

(HIV-1$_{Q23-BG505}$, an R5, clade A recombinant virus) in an infection assay in WT and *CARD8* KO THP-1 cells engineered to express the co-receptor CCR5 (*Figure 3D*). HIV-1$_{Q23-BG505}$ infection also resulted in IL-1β secretion in a CARD8-dependent manner, suggesting that CARD8-dependent inflammasome activation is conserved across HIV-1 strains.

## CARD8-dependent inflammasome activity after HIV-1 infection occurs both early and late in acute infection and depends on the activity of HIV-1$^{PR}$

To gain additional insight into the nature of CARD8 inflammasome responses to HIV-1 infection, we performed a time-course following HIV-1 infection. Unexpectedly, we revealed a statistically significant, CARD8-dependent increase in IL-1β as early as 2 hr after infection (the first timepoint assayed after the initial infection), which plateaued for the next 6 hr and then further increased by 24 hr post-infection (*Figure 4A*). As the early timepoints were sampled prior to reverse-transcription and the genesis of de novo synthesized HIV-1 transcripts (*Mohammadi et al., 2013*), these findings raised the possibility that CARD8 detects the activity of packaged HIV-1$^{PR}$ released into the target cell upon viral entry as well as de novo synthesized Gag-Pol. To test this hypothesis, we treated target cells with the HIV$^{PR}$ inhibitor LPV, which blocked CARD8 inflammasome activation by HIV-1 infection, reinforcing that CARD8 senses HIV-1$^{PR}$ activity. We also considered the possibility that the packaged viral protein R (Vpr) influences CARD8 inflammasome activation. However, we found that an HIV-1 virus lacking Vpr (ΔVpr) also induces CARD8-dependent inflammasome activation (*Figure 4B*). Thus, our findings suggest that HIV-1 infection induces inflammasome activation upon CARD8 detection of HIV-1$^{PR}$ released from the incoming virion as well as newly translated HIV-1$^{PR}$.

## HIV-1 inflammasome activation is dependent on CARD8 sensing of HIV-1$^{PR}$ activity

To further evaluate the role of the human-derived F59-F60 motif of CARD8 after HIV-1 infection, we used a doxycycline (dox)-inducible system to complement *CARD8* KO THP-1 cells with either WT CARD8 or CARD8 cleavage mutants (*Figure 5A*) and probed for subsequent inflammasome activation.

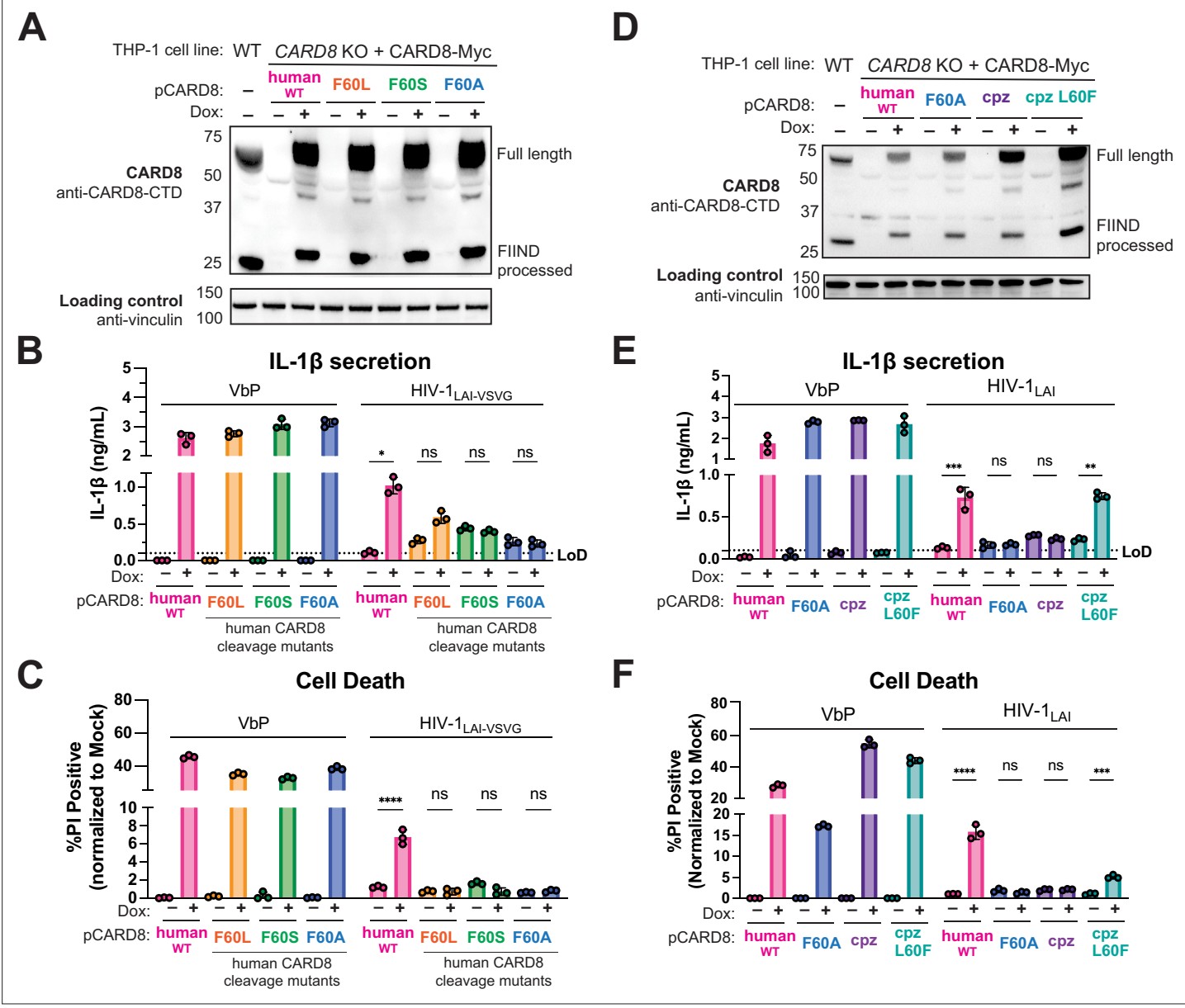

**Figure 5.** HIV-1 inflammasome activation is dependent on a human-specific motif in CARD8. (**A**) *CARD8* knockout (KO) THP-1 lines complemented with different doxycycline (dox)-inducible CARD8 variants (pCARD8) were left uninduced or induced for 18 hr. Immunoblot of wildtype (WT) or complemented *CARD8* KO THP-1 lines treated with ('+') or without ('−') dox was carried out for CARD8 expression using endogenous antibody against CARD8 C-terminal domain (CTD) and loading control (vinculin). FIIND, function-to-find domain. (**B–C**) Complemented *CARD8* KO lines were left uninduced or dox-induced as described in (**A**) and then primed for 4–6 hr with Pam3CSK4 and treated with either 10 μM VbP or HIV-1$_{LAI-VSVG}$ then assessed for (**B**) interleukin (IL)-1β secretion and (**C**) cell death, respectively. (**D**) *CARD8* KO THP-1 lines complemented with different CARD8 variants were left uninduced or induced for 18 hr. Immunoblot of wildtype (WT) or complemented *CARD8* KO THP-1 lines treated with ('+') or without ('−') dox as described in (A). (**E–F**) Complemented *CARD8* KO lines were induced and primed as described in (B) then treated with either 10 μM VbP or HIV-1$_{LAI}$ then assessed for (**E**) IL-1β secretion and (**F**) cell death, respectively. All HIV infections were done at a multiplicity of infection (MOI) such that 30–50% of cells were p24$^{gag}$+ after 24 hr. Dotted line indicates limit of detection (LoD). Datasets represent mean ± SD (n=3 biological replicates). p-Values were determined by two-way ANOVA with Tukey's test using GraphPad Prism 9. ns = not significant, *p<0.05, **p<0.01, ***p<0.001, ****p<0.0001.

The online version of this article includes the following source data and figure supplement(s) for figure 5:

**Source data 1.** Tables of source data for propidium iodide uptake and IL-1β secretion.

**Figure supplement 1.** Human FF motif in chimpanzee CARD8 rescues sensing of HIV/SIV$^{PR}$.

We found that *CARD8* KO THP-1 cells complemented with WT CARD8 underwent IL-1β secretion and cell death in response to both VbP and HIV-1 infection in a dox-dependent manner (*Figure 5B and C*), confirming that HIV inflammasome activation is CARD8-dependent. In parallel, we complemented *CARD8* KO THP-1 cells with the CARD8 cleavage mutants F60L, F60S, F60A. All complemented *CARD8* KO THP-1 cells underwent IL-1β secretion and cell death in response to VbP in dox-treated cells, demonstrating functional CARD8 expression (*Figure 5B and C*). In contrast, infection with both VSV-g pseudotyped (*Figure 5B and C*) and replication competent (*Figure 5E and F*) HIV-1$_{LAI}$ induced IL-1β secretion and cell death only in *CARD8* KO THP-1 cells that were complemented with WT human CARD8, but not CARD8 mutants that are resistant to HIV-1$^{PR}$ cleavage. We also found that *CARD8* KO THP-1 cells complemented with chimpanzee CARD8 restored responsiveness to VbP but not to HIV-1$_{LAI}$ infection, whereas chimpanzee CARD8 L60F (i.e. chimpanzee CARD8 with the phenylalanine at residue 60 as in human CARD8) is cleaved by HIV-1$^{PR}$ and SIVcpz$^{PR}$ (*Figure 5—figure supplement 1*), and can functionally complement human CARD8 responses to HIV-1 infection (*Figure 5D–F*). Thus, human CARD8 detects the enzymatic activity of HIV$^{PR}$ by encoding a motif that functions as a HIV$^{PR}$ substrate, permitting a human-specific CARD8 inflammasome response to HIV-1 infection.

## Discussion

The ability to selectively induce CARD8-dependent pyroptosis in HIV-1 latently infected CD4+ T cells via NNRTI-enforced dimerization of HIV-1$^{PR}$ has garnered much interest as a means to clear the latent reservoir (*Balibar et al., 2023*; *Clark et al., 2022*; *Moore et al., 2022*; *Sparrer and Kirchhoff, 2021*). Here, we demonstrate that CARD8 also senses HIV-1 replication in acutely infected cells, which occurs following HIV-1$^{PR}$ site-specific cleavage of a human-specific motif in the CARD8 tripwire. We further show that the unique motif in human CARD8, which is not present in other sampled hominoids and Old World monkeys, enables its sensing of SIVcpz$^{PR}$ – the precursor to HIV-1, indicating that the precursor viruses to HIV-1 were poised to cleave human CARD8 prior to spillover into humans. These results, along with other recent findings (*Nadkarni et al., 2022*; *Tsu et al., 2023*), demonstrate that CARD8 is a bone fide innate immune sensor of viral infection via sensing viral protease activity, and suggest a model for a human-specific inflammatory response to HIV infection.

### CARD8 as an innate immune sensor of HIV-1$^{PR}$ activity

Some positive-sense RNA viruses that do not package their viral proteases are sensed following de novo synthesized viral protease (*Nadkarni et al., 2022*; *Tsu et al., 2023*). However, as HIV-1 triggers CARD8 inflammasome activation as early as 2 hr post-infection, well before de novo synthesis of viral proteins (*Figure 4A*), our findings suggest that HIV-1 entry is also targeted by CARD8 via the innate immune detection of incoming viral protease activity. This conclusion is supported by a recent report that also found that HIV-1 strains lacking RT or integrase function are still sensed by the CARD8 inflammasome in a manner dependent on HIV-1$^{PR}$ activity (*Wang and Shan, 2023*). Based on these findings, we propose that CARD8 can sense HIV-1$^{PR}$ that is packaged into the virion upon its release into the host cytosol upon viral fusion. This idea is consistent with reports that HIV-1$^{PR}$ can function intracellularly to cleave host targets, and not solely in the context of Gag-Pol dimerization during virion assembly and maturation (*Alvarez et al., 2006*; *Tabler et al., 2022*). We speculate that for HIV-1 this may be particularly relevant for cell-to-cell infection. Thus, our present findings, along with other recent examples of innate immune detection of viral protease activity (*Nadkarni et al., 2022*; *Tsu et al., 2023*), suggest that CARD8's broad antiviral sensing capacity is predicated on its detection of the ubiquitous and essential function of viral proteases, which are evolutionarily constrained by their requirement to target both viral polyprotein and host targets.

We find that inflammasome responses downstream of CARD8 are modulated by TLR stimulation. For example, CARD8-dependent cell death is modestly enhanced by TLR priming by an unknown mechanism (*Figure 3*). On the other hand, IL-1β secretion following HIV-1 infection is strictly dependent on TLR priming, consistent with its established role for transcriptional upregulation of IL-1β (*Chan and Schroder, 2019*). These findings may offer a potential explanation for conflicting reports as to whether or not primary CD4+ T cells undergo pyroptosis and induce IL-1β secretion in response to HIV-1 infection (*Doitsh et al., 2014*; *Muñoz-Arias et al., 2015*). Our findings that several TLR agonists effectively prime CARD8 inflammasome responses (*Figure 3B and C*) suggest that HIV-1

pathogen-associated molecular patterns (i.e. viral nucleic acids) and/or circulating microbial ligands from gut epithelial breakdown, a hallmark of acute HIV-1 disease (*Sandler and Douek, 2012*), are potential sources for priming of HIV-1 target cells in vivo. Moreover, given the exciting potential of combinatorial host- and virus-directed strategies of HIV-1 reservoir clearance by lowering CARD8 activation threshold (via VbP) and enforced HIV$^{PR}$ cytosolic activity (*Balibar et al., 2023*), our findings may also guide therapeutic strategies that leverage HIV$^{PR}$-dependent CARD8 inflammasome activation, which may be bolstered by adjuvants that induce TLR signaling (*Kim and Shan, 2022*; *Moore et al., 2022*; *Wang et al., 2021*).

## Human CARD8 as a maladaptation to HIV-1 infection

Several adaptations of SIVcpz have occurred following its spillover in humans, including Vpu antagonism of human tetherin/Bst2 (*Lim et al., 2010*; *Sauter et al., 2009*), a mutation in MA that allows infection of human tissues (*Bibollet-Ruche et al., 2012*), and the adaptation of Vif to antagonize one of the human polymorphisms in APOBEC3H (*Zhang et al., 2017*). However, other host-virus interactions important for permitting the establishment of the HIV/AIDS (acquired immunodeficiency syndrome) pandemic, such as that of SIVcpz Vif with APOBEC3G, arose in intermediate hosts where no further adaptations were required for passage to humans (*Binning et al., 2019*; *Etienne et al., 2013*). Our findings suggest that the interaction of HIV-1$^{PR}$ with human CARD8 is distinct from these scenarios, as SIVcpz$^{PR}$ already had the capacity to cleave human CARD8 before its cross-species transmission to humans, despite the fact that chimpanzee CARD8 is not itself cleaved by SIVcpz$^{PR}$ due to the lack of the FF motif at amino acid 59/60 (*Figures 1 and 2*).

The F59-F60 motif that confers human CARD8 with the unique capacity to sense HIV/SIVcpz$^{PR}$ is conserved across all humans based on publicly available datasets, as well as being present in a Neanderthal genome, suggesting a genetic sweep occurred in favor of a phenylalanine at position 60. HIV-1 emerged within the past century (*Sharp and Hahn, 2010*) and therefore could not have driven the evolution of the HIV-1$^{PR}$ cleavage site in human CARD8. However, human *CARD8* is highly polymorphic, and multiple residues of the N-terminus of CARD8, including those that allow CARD8 sensing of extant human pathogenic viruses including coronaviruses and picornaviruses, show strong evidence of positive selection, an evolutionary signature consistent with a history of host-pathogen conflict (*Daugherty and Malik, 2012*; *Tsu et al., 2023*). Indeed, the HIV-1$^{PR}$ cleavage site in CARD8 overlaps with a site that is cleaved by the coronavirus 3CL protease (*Tsu et al., 2023*). Although it is possible that the human-specific F60 was fixed stochastically or as a passenger mutation, we favor a scenario in which human CARD8 sensing of HIV-1$^{PR}$ arose as a consequence of CARD8 adaptation to another virus (*Nadkarni et al., 2022*; *Tsu et al., 2023*). Thus, we speculate that an ancient infection of our human ancestors may be responsible for our modern-day maladaptation to HIV-1.

## Possible links to pathogenesis

HIV-1 disease progression to AIDS is characterized by dramatic depletion of CD4+ T cells including via pyroptosis (*Doitsh et al., 2014*) and chronic inflammation accompanied by high levels of plasma cytokines including IL-1 (*Arditi et al., 1991*; *Muema et al., 2020*). As such, multiple inflammasomes have previously been implicated for HIV-dependent inflammasome activation, although the exact mechanisms have remained unclear (*Monroe et al., 2014*; *Zhang et al., 2021*). Here, we show that HIV infection induces pyroptotic cell death and IL-1β secretion via CARD8 recognition of HIV$^{PR}$ activity. Our finding that HIV-1 infection is sufficient to induce inflammasome activation, along with the presence of CARD8 in relevant T cell populations (*Clark et al., 2022*; *Johnson et al., 2020*; *Linder et al., 2020*), also suggests that CARD8 contributes to HIV pathogenesis. Consistent with this hypothesis, recent publications show that HIV-1 infection drives CARD8-dependent pyroptotic cell death both in primary human CD4+ T cells ex vivo and in humanized mouse models of HIV-1 (*Wang and Shan, 2023*). It is also possible that IL-1β release after HIV-1-dependent CARD8 activation after HIV-1 infection could contribute to pathogenesis since IL-1β induces the differentiation of Th17 cells (*Chung et al., 2009*), a highly HIV-susceptible CD4+ T cell subtype, as well as the recruitment of other target immune cells (*Rider et al., 2011*).

SIVs are believed to be generally non-pathogenic in their reservoir hosts with the SIVsmm in sooty mangabeys and SIVagm in African green monkeys as the best studied examples (*Jasinska et al., 2023*). SIVcpz in naturally infected chimpanzees is pathogenic although not to the extent of HIV-1

group M infection of untreated humans (*Keele et al., 2009*). In contrast, SIVs can cause disease in a new species, including experimental SIV infections of macaque monkeys. It is tempting to speculate that these species-specific differences could be, in part, mediated by differential CARD8 inflammasome activation, which in turn influences the extent of CD4+ T cell depletion, chronic immune activation, and bystander cell immunopathology – key pathogenic events that drive the progression to AIDS in the absence of antiretroviral therapy (*Kaur et al., 1998*; *Keele et al., 2009*; *Paiardini and Müller-Trutwin, 2013*). Although our data demonstrates that functional HIV and SIVcpz protease recognition motifs outside of the F59-F60 are absent in human and chimpanzee CARD8, it remains possible that other SIVs have distinct protease specificities that allow for cleavage of species-specific recognition motifs in CARD8 in non-human primates. Indeed, the substrate specificity of SIVmac239[PR] is distinct from HIV-1[PR] (*Figure 2—figure supplement 1*), which may be relevant to CARD8 inflammasome activation and CD4+ T cell depletion in experimental macaque infections. We suggest that future work determining these host- and virus-specific interactions is an important consideration when evaluating HIV pathogenesis in non-human primate models.

## Methods

### Plasmids

psPAX2 and pMD2.G were gifts from Didier Trono (Addgene). The dox-inducible pLKO-puro vector (*Busnadiego et al., 2014*) was a gift from Melissa Kane. Infectious molecular clones for SIVcpz$_{EK505}$ and SIVcpz$_{LB715}$ were gifts from Beatrice Hahn (*Barbian et al., 2015*; *Keele et al., 2006*). HIV-1$_{Q23}$ Δenv provirus and the HIV-1$_{Q23.BG505}$ proviruses were gifts from Julie Overbaugh (*Haddox et al., 2018*; *Poss and Overbaugh, 1999*). HIV-1$_{LAI}$ and HIV-2$_{Rod}$ were previously described (*Guyader et al., 1987*; *Peden et al., 1991*). The HIV-1$_{LAI}$ΔVpr mutant has a frameshift mutation that inactivates the *vpr* gene as described (*Rogel et al., 1995*). CARD8 sequence IDs used for phylogenetic analysis in *Figure 1A* can be found in *Supplementary file 1c*. For CARD8 cleavage assays, the coding sequences of human CARD8 (NCBI accession NP_001171829.1) and chimpanzee CARD8 (NCBI accession XM_024351500.1) were cloned into the pcDNA3.1 backbone (Addgene) with an N-terminal mCherry tag using BamHI and EcoRI cut sites. For dox-inducible complementation assays, the coding sequences of human and chimpanzee CARD8 were cloned into the pLKO-puro backbone using the SfiI site. Point mutations were introduced using overlapping PCR. Full list of primer sequences can be found in *Supplementary file 1a*.

### Cell culture

THP-1 cells (ATCC) were cultured in RPMI (Invitrogen) with 10% FBS, 1% penicillin/streptomycin antibiotics, 10 mM HEPES, 0.11 g/L sodium pyruvate, 4.5 g/L D-glucose and 1% Glutamax. HEK 293T (ATCC) were cultured in DMEM (Invitrogen) with 10% FBS and 1% penicillin/streptomycin antibiotics. All puromycin selections were done at 0.5 µg/mL. For complemented dox-inducible lines, tetracycline-free FBS (Sigma) was used to prevent background CARD8 expression. All lines routinely tested negative for mycoplasma bacteria (Fred Hutch Specimen Processing & Research Cell Bank).

### Immunoblotting

Cells were washed once with 1× PBS before harvesting in NP-40 buffer with protease inhibitor (200 mM NaCl, 50 mM Tris pH 7.4, 0.5% NP-40 alternative, 1 mM dithiothreitol, and Roche Complete Mini, EDTA-free tablets; catalog no. 11836170001). Cytoplasmic lysates were clarified via centrifugation and combined with 4× NuPage LDS Sample Buffer (Invitrogen) containing 10% β-mercaptoethanol and boiled for 5–10 min. Samples were run on a 4–12% SDS-PAGE gel using morpholineethanesulfonic acid buffer, transferred to a nitrocellulose membrane using a Pierce G2 Fast Blotter (Thermo Scientific), blocked in 5% nonfat milk then probed for with primary antibodies diluted in 2.5% milk for mCherry (for CARD8 cleavage), p24[gag] (for HIV[PR] activity), CARD8 C-terminus (for KO validation and complementation), and vinculin (loading control). Blots were washed three times with PBS-T (0.1% Tween-20), incubated with secondary HRP-conjugated antibodies, washed three times again, and then developed with SuperSignal West Femto Maximum Sensitivity Substrate (Fisher Scientific). Further antibody specifications/concentrations and clone info are described in *Supplementary file 1b*.

## CARD8 cleavage assay

HEK293T cells were seeded at $1.5–2 \times 10^5$ cells/well in 24-well plates the day before transfection using TransIT-LT1 reagent at 1.5 µL transfection reagent/well (Mirus Bio LLC). One hundred ng of indicated constructs encoding an N-terminal mCherry-tagged CARD8 were co-transfected into HEK293T cells with either 400 ng of pcDNA3.1 empty vector ('–') or 400 ng of HIV provirus or SIVcpz provirus. HIV Δenv proviruses were used for immunoblots in *Figure 1* and *Figure 2—figure supplement 1*, while infectious HIV and SIVcpz provirus were used for immunoblots in *Figure 2*. Cytoplasmic lysates were harvested 24 hr post-transfection and immunoblotted as described above.

## FLICA assay

Live cells were incubated in media containing FLICA dye (Immunochemistry Technologies, cat. #97) at a dilution of 1:60-1:100 for 30 min at 37°C then washed and fixed according to the manufacturer's protocol. Stained cells were flowed for analysis on a BD Celesta within 18 hr post-staining.

## *CARD8 and CASP1* KO generation

*CARD8* and *CASP1* KO THP-1 cells were generated similarly to *NLRP1* KO described previously (*Tsu et al., 2021a*). Briefly, a *CARD8* or *CASP1*-specific sgRNA was designed using CHOPCHOP (*Labun et al., 2019*), and cloned into a plasmid containing U6-sgRNA-CMV-mCherry-T2A-Cas9 using ligation-independent cloning. THP-1 cells were electroporated using the Bio-Rad Gene Pulser Xcell. After 24 hr, mCherry-positive cells were sorted and plated for cloning by limiting dilution. Monoclonal lines were validated as KO by deep sequencing and OutKnocker analysis, as described previously (*Schmid-Burgk et al., 2014*; *Schmidt et al., 2016*). KO lines were further validated by immunoblot and functional assays. sgRNA used to generate KO are described in *Supplementary file 1a*.

## CCR5+ cell line generation

WT or *CARD8* KO THP-1 cells were transduced with pHIV-CCR5/ZsGreen as previously described (*Montoya et al., 2023*). Cells were sorted 4 days post-transduction on a Sony MA900.

## CARD8 complementation

HEK293T were seeded at $2 \times 10^5$ cells/well in six-well plates the day before transfection using TransIT-LT1 reagent (Mirus Bio LLC) at 5.8 µL transfection reagent/well. Cells were co-transfected with pLKO-CARD8, psPAX2, and pMD2.G and media was replaced the next day. Virus was harvested 2 days post-transfection and underwent one freeze thaw cycle at –80°C before transducing *CARD8* KO THP-1 cells. *CARD8* KO THP-1 cells were seeded at $2 \times 10^5$ cells/well in six-well plates and transduced with 800 µL virus in the presence of 1 µg/mL polybrene via spinoculation at $1100 \times g$ for 30 min at 30°C then puro-selected 24 hr post-transduction.

## HIV-1$_{LAI}$, HIV-1$_{Q23\text{-}BG505}$, and HIV-1$_{LAI\text{-}VSVG}$ production

293T cells were seeded at $2 \times 10^5$ cells/well in six-well plates the day before transfection using TransIT-LT1 reagent (Mirus Bio LLC) at 3 µL transfection reagent/well as previously described (*OhAinle et al., 2018*). For HIV-1 production, 293Ts were transfected with either 1 µg/well HIV$_{LAI}$ proviral DNA or 1 µg/well HIV$_{LAI}$ Δenv DNA and 500 ng/well pMD2.G for HIV-1$_{LAI}$ and HIV-1$_{LAI\text{-}VSVG}$, respectively. One day post-transfection, media was replaced. Two or three days post-transfection, viral supernatants were collected and filtered through a 20 µm filter and aliquots were frozen at –80°C. HIV-1$_{LAI}$ and HIV-1$_{LAI\text{-}VSVG}$ proviruses were previously described (*Bartz and Vodicka, 1997*; *Gummuluru et al., 2003*; *Peden et al., 1991*). HIV-1$_{Q23\text{-}BG505}$ was produced in the same way as HIV-1$_{LAI}$.

## THP-1 priming and HIV-1 infection

THP-1 cells were seeded at $1 \times 10^5$ cells/well in 96-well U-bottom plates in media containing TLR agonist (*Supplementary file 1b*) for 4–6 hr or overnight then treated with either Val-boroPro (10 µM) or nigericin (5 µg/mL) or infected with HIV-1$_{LAI}$, HIV-1$_{Q23\text{-}BG505}$, or HIV-1$_{LAI\text{-}VSVG}$ in the presence of 20 µg/mL DEAE-Dextran via spinoculation at $1100 \times g$ for 30 min at 30°C. All infections were done at an MOI <1. 24 hr post-infection or VbP treatment (2 hr for nigericin), supernatants were collected for IL-1β quantification (see IL-1R reporter assay), and cells were stained with PI or FLICA dye then fixed and stained with $p24^{gag}$-FITC for flow cytometry.

## IL-1R reporter assay

To quantify the IL-1β secretion, HEK-Blue IL-1β reporter cells (Invivogen) were used whereby binding of IL-1β to the surface receptor IL-1R1 results in the downstream activation of NF-kB and subsequent production of secreted embryonic alkaline phosphatase (SEAP) in a dose-dependent manner as previously described (*Tsu et al., 2021a*). SEAP levels were detected using a colorimetric substrate assay, QUANTI-Blue (Invivogen), by measuring an increase in absorbance at OD655. Culture supernatant from treated or infected THP-1 cells was transferred to HEK-Blue IL-1β reporter cells plated in 96-well format in a total volume of 200 μL per well at $5\times10^5$ cells/well. On the same plate, serial dilutions of recombinant human IL-1β (Peprotech) were added to generate a standard curve for each assay. After 24 hr, SEAP levels were assayed by adding 50 μL of the supernatant from HEK-Blue IL-1β reporter cells to 150 μL of QUANTI-Blue colorimetric substrate along with 0.25% Tween-20 to neutralize HIV virions in supernatant before readout. After incubation at 37°C for 15–30 min, absorbance at OD655 was measured on an Epoch Microplate Spectrophotometer (BioTek) and absolute levels of IL-1β were calculated relative to the standard curve.

## Acknowledgements

We thank everyone in the Emerman and Mitchell labs, Amandine Chantharath for assistance with cloning, Abby Felton for assistance with cell sorting, Joy Twentyman for assistance with cell maintenance, Matt Daugherty, Emily Hsieh, and Brian Tsu for critical reading of the manuscript, Janet Young for helpful discussions, Melissa Kane for kindly providing the dox-inducible plasmid (pLKO-puro) used for complementation experiments, Liang Shan and his lab members for discussions and sharing of unpublished results, and the Fred Hutchinson Shared Resources Genomics, Flow Cytometry, and Specimen Processing & Research Cell Bank cores. LPV (HRP-9481) and p24$^{gag}$ antibody (ARP-3537) were provided by the AIDS Reagent Program, Division of AIDS, NIAID, NIH. This work was supported by grants from the National Institutes of Health (NIH) (DP2 AI 154432-01) and the Mallinckrodt Foundation to PSM; NIH grants DP1 DA051110-03 to ME, and University of Washington Cellular and Molecular Biology Training Grant (T32 GM007270) to JK.

## Additional information

### Funding

| Funder | Grant reference number | Author |
|---|---|---|
| National Institute of Allergy and Infectious Diseases | DP2 AI 154432-01 | Patrick S Mitchell |
| Edward Mallinckrodt Jr. Foundation | | Patrick S Mitchell |
| National Institute on Drug Abuse | DP1 DA051110 | Michael Emerman |
| National Institute of General Medical Sciences | T32 GM007270 | Jessie Kulsuptrakul |

The funders had no role in study design, data collection and interpretation, or the decision to submit the work for publication.

### Author contributions

Jessie Kulsuptrakul, Conceptualization, Formal analysis, Validation, Investigation, Visualization, Methodology, Writing – original draft, Writing – review and editing; Elizabeth A Turcotte, Resources, Methodology, Writing – review and editing; Michael Emerman, Conceptualization, Resources, Formal analysis, Supervision, Funding acquisition, Visualization, Methodology, Writing – original draft, Writing – review and editing; Patrick S Mitchell, Conceptualization, Resources, Formal analysis, Supervision, Funding acquisition, Validation, Investigation, Visualization, Methodology, Writing – original draft, Writing – review and editing

## Author ORCIDs

Jessie Kulsuptrakul ⓘ http://orcid.org/0000-0003-3881-4686
Michael Emerman ⓘ http://orcid.org/0000-0002-4181-6335
Patrick S Mitchell ⓘ http://orcid.org/0000-0001-8375-9060

## Decision letter and Author response

Decision letter https://doi.org/10.7554/eLife.84108.sa1
Author response https://doi.org/10.7554/eLife.84108.sa2

---

## Additional files

### Supplementary files

• Supplementary file 1. Primers, reagents, and primate CARD8 sequence info. (a) List of primers, gBlocks, and sgRNA sequences. (b) List of antibodies/reagents. (c) Primate CARD8 gene IDs.

• MDAR checklist

• Source data 1. Uncropped western blots tiffs from *Figures 1–5*.

• Source data 2. Uncropped western blots pdf from *Figures 1–5* .

• Source data 3. Uncropped western blots pdf from *Figure 2—figure supplement 1*, *Figure 3—figure supplement 1*, *Figure 5—figure supplement 1*.

### Data availability

All data generated or analyzed during this study are included as source data files in the article.

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
