## [Editor Report]

Kulsuptrakul and colleagues provide convincing evidence that the human inflammasome-forming sensor CARD8 contains a specific F-F motif that allows cleavage by the proteases of HIV-1 and emerged after separation of chimpanzees and humans. In comparison, CARD8 proteins from non-human primates contain changes in this motif and are largely resistant to proteolytic activation. These important findings suggest a potential role of CARD8 cleavage and inflammasome activation in primate lentiviral pathogenesis.

---

## [Decision Letter]

**Decision letter after peer review:**

Thank you for submitting your article "A human-specific motif facilitates CARD8 inflammasome activation after HIV-1 infection" for consideration by *eLife*. Your article has been reviewed by 3 peer reviewers, and the evaluation has been overseen by Frank Kirchhoff as Reviewing Editor and Reviewer #1, and Miles Davenport as the Senior Editor.

Essential revisions:

1) The potential relevance for pathogenesis remains speculative since effects in primed THP-1 cells infected by VSV-G pseudotyped viruses were modest and no evidence is presented that wild-type SIVcpz or HIV-1 induces inflammatory responses in primary viral target cells in a CARD8-dependent manner. Such evidence would significantly increase the significance of the study and seems critical to warrant conclusions about the potential impact on viral pathogenesis. It should also be clearly mentioned that SIVcpz causes disease in wild chimpanzees (Keele et al., 2009).

2) Conclusions that CARD8 cleavage and inflammasome activation are human-specific should be cautioned or strengthened by further evidence that the simian CARD8 cannot be activated by SIV in relevant target cell types. While the results show that human CARD8 is more sensitive to cleavage by HIV-1 and SIVcpz protease than CPZ CARD8, the possibility that some SIV proteases may cleave CARD8 in their simian hosts cannot be excluded.

*Reviewer #1 (Recommendations for the authors):*

The study is interesting and the data convincingly show that F60 renders human CARD8 sensitive to cleavage by SIVcpz and HIV-1 proteases and is associated with higher levels of cell death and IL1-ß production in THP-1 cells infected with VSV-G pseudotyped HIV-1. The potential relevance for pathogenesis remains speculative since no evidence is presented that wild-type SIVcpz or HIV-1 induces inflammatory responses in primary viral target cells in a CARD8-dependent manner.

Specific points:

Lines 2-3. "Simian immunodeficiency viruses (SIVs) are generally non-pathogenic in their natural hosts". This seems like an overstatement. It has been clearly shown that SIVcpz causes disease in wild chimpanzees (Keele et al., 2009). Also, experimental evidence for lack of disease in natural SIV infection comes mainly just from two of ~40 infected species and rare cases of the disease have even been reported in African green monkeys and sooty mangabeys.

Figure 2B: the levels of EK505 Gag-Pol proteins are pretty low. Is this due to inefficient expression or antibody recognition?

Figure 3: the levels of cell death and IL-1ß secretion are substantially higher for VSV-G-pseudotyped HIV-1 LAI. Do the authors feel that this is just due to higher infection rates or that e.g. different sites of fusion may also contribute to the sensing effect? Do they have an explanation for why the effects of LAI on IL-ß are not dose-dependent (panel B) and why Gag processing seems more effective for wild-type compared to VSV-G-mediated infection in panel C.

In Figures 4E and F, only VSV-G data are shown. Would increase significance if effects can be verified with wild-type HIV-1.

Pg 12, line 27: "relative to HIV-infected patients who quickly progress to AIDS without treatment". Suggest to caution, an average of about 6-8 years doesn't really seem "quick" and is much slower than e.g. progression to simian AIDS in macaques.

Conclusions about the importance of CARD8-dependent inflammasome activation for the pathogenesis should be cautioned throughout or (better) substantiated by further experimental data.

*Reviewer #2 (Recommendations for the authors):*

The authors show a protein alignment in Figure 1A. They need to specify where these data came from or what the species-specific variation is at those positions. The authors claim that only humans have phenylalanine at the 60th position. A more full phylogenetic analysis is needed to support this claim. Additionally, the aggregation of all old-world monkey species needs to be expanded upon in regard to which species are included in this analysis.

In Figure 1B, the 88/89 position was not mentioned as a protease cleavage site in previous studies. Do the authors have any experimental evidence? Otherwise, this should be removed.

Figure 1C: The band present at the 40-45kDa position is explained by the authors as being due to proteasomal degradation but should be validated through the use of proteasome inhibitors or through identifying the exact cleavage site that produces this band.

Figure 1D: There are two bands present at the predicted HIV-PR site which the lower one is present in F60S. Is this band an HIV protease cleavage site as it is present in the HIV-transfected conditions? Does this cleavage lead to functional activation of the CARD8 inflammasome? There is clear evidence of cleavage in the F60L mutant condition, while the authors state that this makes cleavage less efficient. The text should reflect that F60 is not the only amino acid that can be cleaved and should downplay the human specificity due to this cleavage still being present, albeit less efficient. There is also an increased 40-45 kda band size in the mutant conditions, does this increase correspond to alternate cleavage site use, and does the alternative cleavage lead to CARD8 activation?

Figure 2A: There is an additional band at 50kDA present in this blot that was not present in previous blots, the authors should repeat these experiments. There are also faint bands at the SIV/HIV pr cleavage site annotated. The results can be improved by doing biochemical in vitro cleavage experiments using purified CARD8 and viral particles or PR to remove any background that may affect their results.

Figure 3A: These experiments were conducted 24 hours post-infection which can allow for the production of viral proteins. The authors should use earlier time points (<12hrs) or block viral RT or integration to show cell death exclusively caused by the incoming viral proteins as stated in the results. Notably, cell death was observed also in CARD8-KO cells in Figure 4C.

In figure 3B, the authors mention that priming is required for IL-1B release but not cell death. Is it just for THP-1 cells, also correct in primary macrophages?

Can the authors perform similar experiments in Figure 3 using primary CD4 T cells?

In Figures4e and f, the authors also should test whether SIVcpz activates the CARD8cpz, which may have other potential cleavage sites. They should also create other non-human primate species CARD8 proteins to show that this is a human-specific activation as their results can only draw conclusions about human variation at the cleavage site and chimpanzee CARD8 activation.

*Reviewer #3 (Recommendations for the authors):*

1) The conclusion of "…the CARD8 inflammasome as a potential driver of HIV pathogenesis" is partially supported. As described in the summary above, HIV/SIVcpz PR-mediated CARD8 proteolysis at the N-terminus is inefficient to start with (Figures 1 and 2). In primed THP-1 cells, inflammasome activation associated with HIV-1 infection was detectable but much less profound compared to the VbP control.

2) Cell death measured by propidium iodide staining could be caused by multifold factors and thus is a rather generic pathological marker. HIV infection also has a pathological impact independent of PR activity, which is difficult to distinguish with the reported experimental approaches. Accordingly, these factors should be taken into consideration for the interpretation of cell death data.

---

## [Author Response]

Essential revisions:1) The potential relevance for pathogenesis remains speculative since effects in primed THP-1 cells infected by VSV-G pseudotyped viruses were modest and no evidence is presented that wild-type SIVcpz or HIV-1 induces inflammatory responses in primary viral target cells in a CARD8-dependent manner. Such evidence would significantly increase the significance of the study and seems critical to warrant conclusions about the potential impact on viral pathogenesis. It should also be clearly mentioned that SIVcpz causes disease in wild chimpanzees (Keele et al., 2009).

Yes, we agree that the relevance for pathogenesis is speculative. We have extensively rewritten the manuscript to focus on the evolutionary and mechanistic aspects of lentiviral protease-dependent activation of the CARD8 inflammasome that this manuscript provides. We have moved all of the pathogenesis speculation, including clearly stating that SIVcpz causes disease in wild chimpanzees, to the Discussion where it is clearly marked as speculation.

We also point out that other groups have now independently shown that HIV-1 protease induces CARD8 inflammasome activation in primary T cells (Wang et al., 2021, Science; Clark et al. 2022, Nature Chem Biol; Balibar et al. 2023, Science Trans Med; Wang and Shan, 2023, BioRxiv). We now cite these references to contextualize our Discussion.

2) Conclusions that CARD8 cleavage and inflammasome activation are human-specific should be cautioned or strengthened by further evidence that the simian CARD8 cannot be activated by SIV in relevant target cell types. While the results show that human CARD8 is more sensitive to cleavage by HIV-1 and SIVcpz protease than CPZ CARD8, the possibility that some SIV proteases may cleave CARD8 in their simian hosts cannot be excluded.

We have added experiments to address this. Building upon our observation that chimpanzee CARD8 is not cleaved nor activated by HIV-1 or SIVcpz protease, we made a chimpanzee CARD8 L60F mutant, to determine if that single difference between human and chimpanzee CARD8 explains the difference in sensing of HIV-1 and SIVcpz protease. We find that the chimpanzee CARD8 L60F is both cleaved by HIV-1 and SIVcpz protease, and induces inflammasome activation upon HIV-1 infection (Figure 5 and Figure 5 – supplement 1). These data clarify not only that chimpanzee CARD8 is not cleaved or activated at another site, but also demonstrate that the humanspecific F60 on the chimpanzee CARD8 background is sufficient to confer sensing of HIV-1 and SIVcpz protease.

We agree that we cannot argue that no other primate CARD8 is activated by another lentiviral protease since we have not done a broad functional testing of others except for sequence analysis, and therefore we have modified our claims to those that we have tested. We do include data that Neanderthals also encode a CARD8 with the F59-F60 motif (Figure 1A), strengthening our argument that specificity for HIV-1^PR^ evolved prior to the emergence of HIV in humans.

Additionally, in response to reviewer requests, we added a great deal of additional experiments to strengthen the mechanistic aspects of this manuscript, including:

– Multiple TLR agonists augment HIV-1 induced cell death and are required for HIV-1 induced IL-1 (Figure 3B and 3C).

– WT HIV-1 *infection* induces CASP1 activation in a CARD8-dependent manner (Figure 3D). In addition, we also now show that replication-competent HIV-1 (similar to VSVg pseudotyped virus) also induces CARD8 inflammasome activation in a manner dependent on cleavage of CARD8 (Figure 5D-F), as well as function of the HIV-1^PR^ using the protease inhibitor Lopinavir (Figure 4B).

– Infection with a clinical HIV-1 strain that utilizes a distinct co-receptor (CCR5) relative to LAI, which also activates the CARD8 inflammasome (Figure 3E).

– The surprising finding that CARD8 activation occurs as early as 2h post-infection, suggesting that HIV-1^PR^ released from the virion into target cells is sensed by CARD8 immediately following viral entry (Figure 4).

As such, we now emphasize the evolution-focused nature of our study that reveals the origins of HIV-1 sensing by human CARD8, which we believe also contributes important mechanistic insights about human CARD8 inflammasome activation by HIV-1 infection.

Reviewer #1 (Recommendations for the authors):The study is interesting and the data convincingly show that F60 renders human CARD8 sensitive to cleavage by SIVcpz and HIV-1 proteases and is associated with higher levels of cell death and IL1-ß production in THP-1 cells infected with VSV-G pseudotyped HIV-1. The potential relevance for pathogenesis remains speculative since no evidence is presented that wild-type SIVcpz or HIV-1 induces inflammatory responses in primary viral target cells in a CARD8-dependent manner.Specific points:Lines 2-3. "Simian immunodeficiency viruses (SIVs) are generally non-pathogenic in their natural hosts". This seems like an overstatement. It has been clearly shown that SIVcpz causes disease in wild chimpanzees (Keele et al., 2009). Also, experimental evidence for lack of disease in natural SIV infection comes mainly just from two of ~40 infected species and rare cases of the disease have even been reported in African green monkeys and sooty mangabeys.

We have moved the parts about possible links to pathogenesis from the Introduction to the Discussion where it is clearly marked as speculation. We have revised the text to be more precise about where the lack of disease has been observed, and now cite Keele et al., 2009 as suggested.

Figure 2B: the levels of EK505 Gag-Pol proteins are pretty low. Is this due to inefficient expression or antibody recognition?

The HIV-1 gag monoclonal antibody poorly recognizes the EK505 gag, and some of the other lentiviral gags we have now tested (e.g., SIVmac239; Figure 2 – Supplement 1). There is, however, enough cross-reactivity to be able to show that the gag polyprotein is cleaved. We now note this in the figure legends.

Figure 3: the levels of cell death and IL-1ß secretion are substantially higher for VSV-G-pseudotyped HIV-1 LAI. Do the authors feel that this is just due to higher infection rates or that e.g. different sites of fusion may also contribute to the sensing effect? Do they have an explanation for why the effects of LAI on IL-ß are not dose-dependent (panel B) and why Gag processing seems more effective for wild-type compared to VSV-G-mediated infection in panel C.

This is due to higher infection levels with the pseudotyped virus. We have now repeated these experiments with equivalent infectious doses of virus and find that the levels of IL-1β and cell death are the same whether or not viral entry occurs via WT envelope or VSV-G (e.g., Figure 3C and Figure 5E and 5F).

We have not experimentally addressed the lack of dose-dependency of IL-1β. We speculate that even at lower infectivity levels that cells release a large bolus of IL-1β that may confound detection of dose-responsiveness. It has also been shown that inflammasome activation and IL-1β processing and release can occur in the absence of cell death (e.g,. Evavold et al., 2018, Immunity), which may also contribute to the lack of observed dose-dependency. We agree that this is an interesting observation, but is not an aspect that we have further investigated in the present manuscript. We have opted to replace this data with what we feel is a more informative panel showing that multiple TLR agonists are capable of enhancing HIV-1 induced cell death and are required for HIV-1 induced IL-1 cytokine release (Figure 3C).

In Figures 4E and F, only VSV-G data are shown. Would increase significance if effects can be verified with wild-type HIV-1.

We have now repeated this with WT infectious HIV-1 LAI and observe a similar result to the VSV-g pseudotyped HIV-1 LAI (Figure 5E-5F).

Pg 12, line 27: "relative to HIV-infected patients who quickly progress to AIDS without treatment". Suggest to caution, an average of about 6-8 years doesn't really seem "quick" and is much slower than e.g. progression to simian AIDS in macaques.

We have removed this text.

Conclusions about the importance of CARD8-dependent inflammasome activation for the pathogenesis should be cautioned throughout or (better) substantiated by further experimental data.

We have moved the pathogenesis section to the Discussion.

Reviewer #2 (Recommendations for the authors):The authors show a protein alignment in Figure 1A. They need to specify where these data came from or what the species-specific variation is at those positions. The authors claim that only humans have phenylalanine at the 60th position. A more full phylogenetic analysis is needed to support this claim. Additionally, the aggregation of all old-world monkey species needs to be expanded upon in regard to which species are included in this analysis.

We have now added in all sequences from publicly available genomes from Old world monkey nonhuman primates that unambiguously encode an intact CARD8 gene. We have also included Neanderthal CARD8. Accession numbers for these sequences have been added to Supplement Table S3.

We of course agree that other primates for which we do not have a CARD8 sequence could have a phenylalanine at the analogous position to the human F60, and have altered the text to reflect this.

In Figure 1B, the 88/89 position was not mentioned as a protease cleavage site in previous studies. Do the authors have any experimental evidence? Otherwise, this should be removed.

This was our error and the figure has now been modified to remove mention of position 88/89 as a known cleavage site.

Figure 1C: The band present at the 40-45kDa position is explained by the authors as being due to proteasomal degradation but should be validated through the use of proteasome inhibitors or through identifying the exact cleavage site that produces this band.

The cleavage that gives rise to the ~40-45 kDa band was shown by the Bachovchin lab (Hsiao et al. 2022, JBC) to be dependent on the 20S proteasome. The authors also demonstrated that the 40-45 kDa CARD8 product is non-functional. We now explicitly mention this in the text (line 103-104 and figure legends).

Figure 1D: There are two bands present at the predicted HIV-PR site which the lower one is present in F60S. Is this band an HIV protease cleavage site as it is present in the HIV-transfected conditions? Does this cleavage lead to functional activation of the CARD8 inflammasome? There is clear evidence of cleavage in the F60L mutant condition, while the authors state that this makes cleavage less efficient. The text should reflect that F60 is not the only amino acid that can be cleaved and should downplay the human specificity due to this cleavage still being present, albeit less efficient. There is also an increased 40-45 kda band size in the mutant conditions, does this increase correspond to alternate cleavage site use, and does the alternative cleavage lead to CARD8 activation?

The issue of other minor cleavages is addressed with our functional assays in Figure 5 where we complement the *CARD8* KO THP-1 cells with different CARD8 variants. We observe inflammasome activation after HIV-1 infection upon complementation with WT human CARD8 and chimpanzee CARD8 L60F in which amino acid 60 has been changed to the human phenylalanine amino acid (Figure 5D-5F). Thus, whereas over-expression of gagpol may be capable of spurious CARD8 cleavage, this does not occur during HIV-1 infection. We also point out that not all cleavage events in the CARD8 N-terminus, or the analogous NLRP1 N-terminus, are activating (e.g., Sandstrom et al., 2019, Science, Hsiao et al. 2022, JBC, Wang et al., 2021, Science).

Figure 2A: There is an additional band at 50kDA present in this blot that was not present in previous blots, the authors should repeat these experiments. There are also faint bands at the SIV/HIV pr cleavage site annotated. The results can be improved by doing biochemical in vitro cleavage experiments using purified CARD8 and viral particles or PR to remove any background that may affect their results.

As stated above, the issue of other minor cleavages is addressed with our functional assays in Figure 5 where we complement *CARD8* KO THP-1 cells with different CARD8 variants. As such, we do not believe that using purified CARD8 in biochemical assays would substantially improve our interpretations, and instead favor conclusions based on the cumulative evidence across cleavage and functional assays, which are consistent with specific HIV-1 protease cleavage at the F59-F60 motif.

Figure 3A: These experiments were conducted 24 hours post-infection which can allow for the production of viral proteins. The authors should use earlier time points (<12hrs) or block viral RT or integration to show cell death exclusively caused by the incoming viral proteins as stated in the results. Notably, cell death was observed also in CARD8-KO cells in Figure 4C.

We thank the reviewer for this insightful suggestion. We have now completed a series of experiments to address the timing of HIV-1 activation of the CARD8 inflammasome. Remarkably, we find that HIV-1 leads to CARD8 inflammasome activation within 2h of infection (Figure 4A), suggesting that the protease packaged within the incoming virus, in addition to de novo translated protease, is sensed by CARD8 upon HIV-1 entry. Consistent with this possibility, a preprint by Wang et al. shows that infection with a RT or integrase mutant virus, or treated cells with a reverse transcriptase inhibitor to block replication prior to the translation of HIV-1 proteins, still leads to CARD8 inflammasome activation (Wang et al. 2023, BioRxiv). We also confirm that CARD8 inflammasome activation is dependent on sensing HIV-1^PR^ activity, as the HIV-1^PR^ inhibitor treatment blocked HIV-1-induced inflammasome activation (Figure 4B).

As Vpr/Vpx are currently the only viral proteins that are thought to be packaged into the HIV virion that remain active to modulate host cell functions in the target cell, we also considered that Vpr may somehow be involved in CARD8 inflammasome activation by HIV-1. We found that HIV-1 mutant strain that lacks Vpr still results in IL-1β secretion from THP-1 WT but not *CARD8* KO cells (Figure 4B).

Taken together, these experiments clearly show that the packaged, incoming protease activates the CARD8 inflammasome. Work recently posted to BioRxiv by Liang Shan’s lab are in agreement with these findings (Wang and Shan, 2023, BioRxiv). These results have been added to the manuscript as a new Figure 4.

As noted above, we also now include a FLICA assay (which monitors CASP1 activation) and matched cytokine and cell death readouts to the panel in Figure 3D. Our results clearly show that CASP1 is activated only in the presence of CARD8 after HIV-1 infection.

In figure 3B, the authors mention that priming is required for IL-1B release but not cell death. Is it just for THP-1 cells, also correct in primary macrophages?

We have not done this experiment in primary macrophages. We would prefer to more fully explore the roles of different primary cell subsets in a subsequent study while this study focuses on the evolutionary and mechanistic consequences of CARD8 variation. We do cite other manuscripts (Clark et al. 2022, Nature Chem Biol; Balibar et al. 2023, Science Trans Med; Wang and Shan, 2023, BioRxiv) that look at CARD8-dependent cleavage by HIV-1 protease in primary T cells.

Can the authors perform similar experiments in Figure 3 using primary CD4 T cells?In Figures4e and f, the authors also should test whether SIVcpz activates the CARD8cpz, which may have other potential cleavage sites. They should also create other non-human primate species CARD8 proteins to show that this is a human-specific activation as their results can only draw conclusions about human variation at the cleavage site and chimpanzee CARD8 activation.

We now have addressed this issue with the cleavage assays. We found that SIVcpz does not cleave chimpanzee CARD8, whereas a chimpanzee CARD8 L60F variant that contains the HIV-1 cleavage motif found in human CARD8, is. This data is now in Figure 5 – Supplement 1.

In addition, we show that SIVmac239 protease can cleave human CARD8 in a manner that depends on the amino acid identity at position 60 (Figure 2 – Supplement 1). We have not generated additional non-human primate CARD8 proteins as we prefer to focus on the evolutionary transition that occurred in the great ape lineage since the common ancestor of chimpanzees and humans that led to the human mutation in CARD8. We agree that characterization of simian lentivirus interactions across non-human primate CARD8s is a compelling next step. However, we feel to do this properly in a way that sufficiently samples diversity on both the virus and host side is a significant amount of work that should be its own story.

Reviewer #3 (Recommendations for the authors):1) The conclusion of "…the CARD8 inflammasome as a potential driver of HIV pathogenesis" is partially supported. As described in the summary above, HIV/SIVcpz PR-mediated CARD8 proteolysis at the N-terminus is inefficient to start with (Figures 1 and 2). In primed THP-1 cells, inflammasome activation associated with HIV-1 infection was detectable but much less profound compared to the VbP control.

In our view, it is not unexpected that VbP would induce an elevated response relative to our viral infection using a MOI<1. We note that the apparent discrepancy between VbP- and HIV-induced CARD8 inflammasome activation is diminished in our new Figure 3D, where infectivity is routinely 30-50% relative to the experiment from the prior figure panel in which infectivity was 4%, 8%, and 45%. Based on our new data on CARD8 sensing of packaged HIV-1^PR^ (Figure 4) we suggest that MOI may be an important determinant of the magnitude of the CARD8 response to HIV-1 infection, which we plan to explore in future studies.

2) Cell death measured by propidium iodide staining could be caused by multifold factors and thus is a rather generic pathological marker. HIV infection also has a pathological impact independent of PR activity, which is difficult to distinguish with the reported experimental approaches. Accordingly, these factors should be taken into consideration for the interpretation of cell death data.

As noted above, we have now more specifically measured CASP1 activation using a FLICA assay (which specifically reports on CASP1 activation) in WT, *CARD8* KO and *CASP1* KO THP-1 cells (Figure 3D). In experiments with both VSV-g pseudotyped and infectious virus, we observed increased FLICA signal in WT but not *CASP1* KO THP-1 cells. Moreover, the FLICA signal in *CARD8* KO THP-1 cells was indistinguishable from the *CASP1* KO THP-1 cells. Thus, our results are consistent with HIV-1 infection inducing CASP1-dependent pyroptosis downstream of CARD8.